# Sunlight-Driven Photocatalytic Active Fabrics through Immobilization of Functionalized Doped Titania Nanoparticles

**DOI:** 10.3390/polym15132775

**Published:** 2023-06-22

**Authors:** Ume Arfa, Mubark Alshareef, Nimra Nadeem, Amjed Javid, Yasir Nawab, Khaled F. Alshammari, Usman Zubair

**Affiliations:** 1Department of Textile Engineering, School of Engineering and Technology, National Textile University, Faisalabad 37610, Pakistannimranadeem692@gmail.com (N.N.); amjad1238@gmail.com (A.J.);; 2Department of Chemistry, Faculty of Applied Science, Umm Al Qura University, Makkah 24230, Saudi Arabia; 3Department of Criminal Justice and Forensics, King Fahad Security College, Riyadh 11461, Saudi Arabia

**Keywords:** photocatalysis, zinc doped TiO_2_, self-cleaning fabrics, dye degradation

## Abstract

Frequent washing of textiles poses a serious hazard to the ecosystem, owing to the discharge of harmful effluents and the release of microfibers. On one side, the harmful effluents from detergents are endangering marine biota, while on the other end, microplastics are observed even in breastfeeding milk. This work proposes the development of sunlight-driven cleaning and antibacterial comfort fabrics by immobilizing functionalized Zn-doped TiO_2_ nanoparticles. The research was implemented to limit the use of various detergents and chemicals for stain removal. A facile sol–gel method has opted for the fabrication of pristine and Zn-doped TiO_2_ nanoparticles at three different mole percentages of Zn. The nanoparticles were successfully functionalized and immobilized on cotton fabric using silane coupling agents via pad–dry–cure treatment. As-obtained fabrics were characterized by their surface morphologies, availability of chemical functionalities, and crystallinity. The sunlight-assisted degradation potential of as-functionalized fabrics was evaluated against selected pollutants (eight commercial dyes). The 95–98% degradation of dyes from the functionalized fabric surface was achieved within 3 h of sunlight exposure, estimated by color strength analysis with an equivalent exposition of bactericidal activities. The treated fabrics also preserved their comfort and mechanical properties. The radical trapping experiment was performed to confirm the key radicals responsible for dye degradation, and h^+^ ions were found to be the most influencing species. The reaction pathway followed the first order kinetic model with rate constant values of 0.0087 min^−1^ and 0.0131 min^−1^ for MB and MO dyes, respectively.

## 1. Introduction

Textiles are often regarded as a second skin to humans. They decorate and protect human bodies while bringing comfort into our lives [1]. There are numerous techniques to wash off the impurities from the clothes, but most of them involve the use of surfactants and water. Water quality testing reveals that surfactants increase biochemical oxygen demand (BOD), chemical oxygen demand (COD), salinity, turbidity, electrical conductivity (EC), nitrogen, nitrate, phosphate, total suspended substances (TSS), and total dissolved substances (TDS) [2]. The situation is deteriorating, owing to introduction of the massive detergency chemicals into freshwater bodies on an annual basis. Besides detergents, the dyestuff is another major contributor to water contamination, posing a crucial threat to biota [3]. As a consequence, the mutagenic and carcinogenic effects of these chemicals found in wastewater may result in nervous system disorders, an improper reproductive system, and liver, kidney, and brain malfunctioning, among other things, for human beings [4]. Another impact of chemicals in water is that they restrict sunlight penetration to aquatic life and cause disturbances in the aquatic environment. On the other hand, frequent washes result in the release of microplastics in the form of microfibers on each wash, for the clothes developed from the intimate blending of cellulosic and synthetic staple fibers. The severity of the situation is such that a study conducted in Italy revealed the presence of microplastics in human breastmilk [5].

Photocatalytic active fabrics have been fostered to reduce the use of surfactants, as these fabrics utilize the potential of light for cleaning without the use of surfactants [6]. There is a universal commercial market for self-cleaning textiles, i.e., self-cleaning cotton fabrics with durability for 36–50 washing cycles that are a class of new product categorized as intelligent fabrics, where fourteen million meters of this fabric is requested just by the European Union market per five years [7]. It was determined that a fabric finished with titania NPs could provide effective protection against bacteria and the discoloration of stains, due to the photocatalytic property of TiO_2_ NPs, as well as provide antimicrobial and UV blocking activities. It is cheap, stable, nontoxic, and biocompatible. It is worth mentioning that self-cleaning textiles have many applications in different fields, such as indoor–outdoor upholstery, the military, agriculture, industrial clothing, carpets, tents, and fabric filters. Self-cleaning finishing has properties that are suitable for use in apparel and home textile applications [1].

TiO_2_ is a wide bandgap semiconductor. In nature, TiO_2_ is usually found in three different crystalline structures: rutile, anatase, and brookite. TiO_2_ in anatase form is the most widespread photocatalyst for hydrogen evolution [8]. However, it cannot be used in the spectrum of visible light, since its bandgap for different crystalline phases (anatase-3.2 eV, rutile-3.0 eV, and brookite-3.3 eV) is in the UV region [9]. Due to the wide bandgap (3.2 eV) of the material, the capacity to absorb solar irradiation and indoor light is only a small fraction (5% or less). Thus, any increase in the photocatalytic efficiency of TiO_2_ achieved by moving its optical response to the visible range would have a significant beneficial impact [10]. As photocatalytic degradation occurs mostly on TiO_2_ surfaces, mass transfer limitations must be reduced. However, the low affinity of TiO_2_ for organic pollutants (particularly hydrophobic contaminants) results in inactive photocatalytic breakdown rates. The instability of TiO_2_ nanoparticles may cause aggregation during photocatalytic degradation, reducing light incidence on active centers, and therefore a reduction in catalytic activity [11,12]. However, small particles may show higher scattering, which can reduce their photocatalytic activity compared to large ones [13].

To overcome all the limitations of TiO_2_-based photocatalysis, the following countermeasures have been adopted in previous studies: (a) modification of the TiO_2_ catalyst to achieve the utilization of visible light [14,15]; (b) optimization of the catalyst synthesis to obtain catalysts with a defined crystal structure, small particle sizes, and high affinity to various organic pollutants [16,17]; and (c) design and development of the second generation of TiO_2_ catalyst with high separation ability, which can be recovered and regenerated effectively [18,19].

Modification of TiO_2_ with transition metals was presented as a successful and cost-effective alternative to increasing its efficiency as a photocatalyst. The transition-metal (TM) ions have multiple valences with an unfilled d-electron structure with the ability to accommodate more electrons. Insertion of TM (by doping) generates heterojunction in the band gap of TiO_2_ results in preventing the recombination of photogenerated electron hole pairs with improved photo catalytic performance in the visible light range [20]. Doping with Zr and Cu, the bandgap of Cu-doped TiO_2_ is 3.7 eV and Zr-doped TiO_2_ was 3.5 eV, and it revealed that the increase in bandgap did not affect the photocatalytic degradation of the dye. A Zn-doped TiO_2_ catalyst showed stronger and broader absorption between 400 nm and 700 nm, indicating that the catalyst effectively used visible light irradiation. A Zn-doped TiO_2_ leads to a significant effect on the absorption of visible light and better photocatalytic activity. This revealed that the bandgap shifted from 3.2 eV for pure TiO_2_ to 2.8 eV for Zn-doped TiO_2_. The redshift was obtained for the transition metal doped TiO_2_ due to the charge transfer that occurred between the impurity bands and the TiO_2_ conduction band [21]. Zn-doped TiO_2_ has been proven to be an effective catalyst because of its corrosion, hardness, semiconducting and magnetic properties, good transparency, and electron mobility. Moreover, doping Zn may facilitate the formation of surface-bound OHs on the surface of TiO_2_ photocatalysts, and significantly increase the surface area of TiO_2_ [22].

This paper was oriented to develop a photocatalytically active fabric using zinc-doped TiO_2_ nanoparticles. The nanoparticles were fabricated using the facile sol–gel method and immobilized on cotton fabric using γ-glycidoxy propyl trimethoxy silane (GPTS) as a coupling agent. The pristine NPs and functionalized fabrics were characterized using FTIR, SEM, and XRD analysis. The functionalized fabrics were subjected to comfort analysis in terms of air and water permeability and tensile strength. The color strength (K/S) of dye-treated functionalized fabrics was estimated, and the percentage reduction in dyes was calculated. The functionalized Zn-doped TiO_2_ nanoparticles have been successfully immobilized on the surface of fabrics with remarkable photodegradation potential under sunlight.

## 2. Materials and Methods

### 2.1. Chemicals and Materials

All the chemicals used for this project were of analytical grade and utilized without further purification. Titanium isopropoxide (97%) was purchased from Sigma-Aldrich, zinc nitrate hexahydrate (Zn(NO_3_)_2_.6H_2_O > 98.0%) was obtained from Daejung, 2-propanol was obtained from EMSURE (99%), ethanol (AR grade) from ACILabscan, polyethylene glycol (1000 BioChemica) from AppliChem, and 3-glycidoxypropyl trimethoxy silane (GPTS) was purchased from Sigma-Aldrich. The mercerized 100% plain cotton fabric having an areal density of 131 g/cm^2^ was purchased from the local textile industry. Deionized water was used throughout the study. Five commercial dyes were chosen as target pollutants, and obtained from Archroma.

### 2.2. Synthesis of NPs and Immobilization on Fabric

#### 2.2.1. Synthesis of TiO_2_ NPs

Pristine TiO_2_ NPs were synthesized using the sol–gel method. For this, two solutions were prepared. Solution 1: titanium isopropoxide (20 mL of 50 mmol.) was added into a solution containing 160 and 20 mL ethanol and isopropanol (i.e., 8:1 *v*/*v*), respectively, under ultrasonication. Solution 2: 2 g cetyltrimethylammonium bromide (CTAB) was added into 100 mL deionized water. Afterward, solution 1 was added to solution 2 slowly to form white ppts. Then, the system was kept under constant stirring for 4 h at 80 °C. Afterward, the solution was heated in a water bath under constant stirring to remove excess water. The obtained precursors were dried at 110 °C for 12 h and calcined at 500 °C for 4 h in a muffle furnace.

#### 2.2.2. Synthesis of Zn@TiO_2_ NPs

Zn@TiO_2_ NPs were synthesized using the protocol described above, except for the addition of zinc nitrate nonahydrate into solution 2. The three different mol ratios (0.5, 1, and 2 mol%) of Zn doping were selected and labeled as Zn@TiO_2_-0.5, Zn@TiO_2_-1, and Zn@TiO_2_-2. The NPs were calcined at 500 °C for 4 h in a muffle furnace.

#### 2.2.3. Functionalization of Pristine TiO_2_ and Zn@TiO_2_ NPs Using GPTS

For the functionalization of NPs, the following procedure was opted. Different percentages of Zn@TiO_2_ NPs (4, 6, and 8%) on the weight of fabric (o.w.f.) were dispersed into 100 mL of isopropanol and ultrasonicated for 1 h. After this, different concentrations of GPTS (75 and 125%) on the weight of NPs (o.w.n.) were added, followed by the adjustment of solution pH at 4.5 using 0.01 M HCl. The solution was refluxed afterward at 60 °C for 4 h. The GPTS (75% o.w.n.) functionalized nanoparticles were labeled as Zn@TiO_2_(4), Zn@TiO_2_(6), and Zn@TiO_2_(8), depending upon the wt% of NPs o.w.f.

#### 2.2.4. Immobilization of Pristine TiO_2_ and Zn@TiO_2_ NPs on Fabric

Before the immobilization of functionalized NPs, the cotton fabric was dipped into a 0.5 M (200 mL) solution of NaOH. This slight alkaline treatment to cotton fabric improves the interaction of the fabric with GPTS functionalized NPs. Immobilization of functionalized NPs was performed by dipping the fabric into the GPTS functionalized NPs solution and rolling the fabric with a padder machine, then the fabric was dried at 100 °C and cured at 150 °C for 3 min in a stenter frame (pin TC-M-28). The samples containing NPs immobilized on the cotton fabric were labeled as C-Zn@TiO_2_(4), C-Zn@TiO_2_(6), and C-Zn@TiO_2_(8). Immobilization of Zn@TiO_2_ NPs on the cellulose-based substrate follows the following steps: Initially, the hydrolysis of silane coupling agent, i.e., GPT, is carried out in the presence of isopropanol. The hydrolyzed methoxy groups of GPTS react further with the OH groups on Zn@TiO_2_ NPs via a condensation reaction. The silane group of GPTS is attached to the NPs, and the epoxy part is bonded to the cellulosic group of cotton; therefore, GPTS act as a bridge between NPs and cotton providing high durability of functionalized fabrics. The as-prepared dispersion when applied on the cotton fabric reacts with the OH groups of cellulose leading to the effective anchoring of doped TiO_2_ NPs on the fabric surface. The detailed reaction mechanism is presented in Figure 1.

### 2.3. Testing and Characterization

The identification of functional groups in TiO_2_, Zn@TiO_2_ NPs, and functionalized fabrics was analyzed by Fourier-transform infrared spectroscopy (FTIR, Perkin Elmer Oswestry Shropshire UK) in the transmittance mode. The surface morphology of pristine cotton fabrics and Zn@TiO_2_ functionalized cotton fabrics were analyzed using a scanning electron microscope (SEM, JSM 6490) with an accelerating voltage of 20 V. The structural phases in nanoparticles were confirmed by XRD analysis via X-ray powder diffractometer, having Cu kα as a radiation source (λ = 0.154 nm). The average crystallite size of as-prepared nanoparticles was calculated from the XRD profile using the Scherrer equation (Equation (1)).
(1)$$D=\frac{k\lambda }{\beta \cos\theta }$$
where *D* represents average crystallite size, λ denoted X-ray wavelength (0.154 nm), K is Scherrer constant, *θ* is Bragg angle, and *β* shows full-width half maximum (FWHM). The K/S (color strength) of all the fabric samples (coated (C), uncoated (UC), both dipped in dyes and dried under dark (*D*), and sunlight (S)) was measured by a spectrophotometer GretagMacbeth 700. The K/S max (color strength) of the dyed sample was calculated using the Kubelkae–Munk equation (Equation (2)).
(2)$$K/S={\left(1-R\right)}^{2}/2R$$
where K is the absorption coefficient, S is the scattering coefficient, and R is the reflectance of the colored sample. The fabric’s comfort analysis, in terms of air and water permeability and tensile strength, was estimated. The antimicrobial activity test of uncoated and coated cotton fabrics was performed by ISO 20743:2013 Transfer Method [23]. All the fabric samples were placed in nutrient agar plates. Before this, the plates were inoculated using 1 mL bacterial culture (inoculum of CFU/mL). The fabrics were pressed for 60 s under a weight of 200 g. The washing durability of functionalized fabrics was tested using ISO 105 CO3 method, and functional properties were examined after washing.

### 2.4. Photocatalytic Activity Test

The pristine Zn@TiO_2_ NPs, GPTS functionalized Zn@TiO_2_ NPs and Zn@TiO_2_ NPs immobilized on the cotton fabric were tested for photocatalytic activity using five dyes (methylene blue, methyl orange, drimaren ultimate yellow, drimaren ultimate red, and drimaren ultimate blue) as model pollutants. For this, different ppm solutions of selected dyes were taken under different conditions of solution pH (acidic, basic, and neutral) and NPs (30 mg), functionalized NPs (1 mL), and functionalized cotton fabric (22 cm) were dipped into 100 mL dye solution. The reaction was kept under ambient sunlight for a specific reaction time. The sunlight intensity was measured using a lux meter. The absorbances of treated and untreated dye solutions at their specific absorption maxima (i.e., λmax = 664 nm, 446 nm, 415 nm, 542 nm, and 600 nm for MB, MO, reactive yellow (Y), reactive red (R), and navy blue (NB), respectively) were taken using UV-visible spectrophotometer (CECIL CE 7200). The treated functionalized and reference fabrics were air-dried in the dark as well as under sunlight, and their K/S values were tested.

### 2.5. K/S (Color Strength) Analysis

The color strength of uncoated and coated fabrics with variable percentages of NPs was tested by dipping the respective cotton fabric into the dye solution (50 ppm for all dyes). After dipping, one portion of the fabric was placed in the dark, and a second portion was placed under ambient sunlight to observe the self-cleaning activity of functionalized fabrics. After exposure for 3 h, the fabrics were subjected to K/S analysis.

### 2.6. Kinetic Study

First and second order kinetic reaction models were selected to quantify the photocatalytic degradation of MB and MO dye. The expressions for the first and second order kinetic models are presented in Equations (3) and (4), respectively [24,25,26].
(3)$$\ln\frac{C_{o}}{C_{t}}=k_{1}.t$$
(4)$$\frac{1}{C_{t}}-\frac{1}{C_{o}}=k_{2}.t$$

Here, K_1_ is the first order rate constant while C_0_ and C_t_ are the dye concentrations at zero and specific reaction time t, respectively [27]. K_1_ and K_2_ are the rate constants for the first and second order kinetic model [28].

## 3. Results and Discussions

### 3.1. Characterization

The SEM micrographs of pristine Zn@TiO_2_-0.5 nanoparticles, Zn@TiO_2_-0.5 functionalized cotton fabric, and functionalized cotton fabric after 10 washes, are presented in Figure 2. The results illustrate the even-shaped Zn@TiO_2_-0.5 particles with a homogeneous distribution. The functionalized cotton fabric exhibits microscale fibers with characteristic parallel ridges. High-resolution images present a clear distribution of nanoparticles throughout the parallel ridges of cotton fabric, generating highly rough surfaces. The uneven surfaces generated at the nanoscale after the functionalization of cotton fabric are advantageous for an effective increase in adsorption surface area. Additionally, the functionalized fabric retained an effective concentration of NPs even after 10 washes, exhibiting apparent anchoring of NPs on cotton fabric via silane coupling agent, i.e., GPTS. Figure 2d,e represents the macroscopic images of uncoated and coated fabrics, respectively. It can be observed that there is no change in the physical appearance of fabric.

The X-ray diffraction patterns of pristine TiO_2_ and Zn-doped TiO_2_ (0.5 mol%, 1 mol%, and 2 mol%) NPs are presented in Figure 3. The strong and sharp diffraction peak represents the high crystallinity of fabricated NPs. The pattern of pristine TiO_2_ showed anatase phase as diffraction peaks at 2ϴ = 25.2°, 37.9°, 48.0°, 53.8°, 55.0°, 62.8°, and 70.1° were observed, which correspond to the crystal planes of (101), (004), (105), (211), (204), (126), and (220) (JCPDS 89-4921). No additional peaks in Zn-doped TiO_2_ NPs were observed, indicating no effective change in the structure of tetragonal TiO_2_ after Zn doping. This could be due to the similarity in ionic sizes of Ti^4+^ and Zn^2+^. However, a small reduction in the anatase phase peak intensities was observed after Zn doping, owing to the formation of ZnTiO_3_ species at higher Zn loading. Therefore, increasing the Zn doping promotes the formation of ZnTiO_3_, which results in anatase to rutile phase transformation. The crystallite size of NPs was calculated to be 10.4 nm, 8.8 nm, 6.8 nm, and 5.6 nm for TiO_2_, Zn@TiO_2_-0.5, Zn@TiO_2_-1, and Zn@TiO_2_-2, respectively, using the Scherrer formula.

The chemical structure of pristine NPs (i.e., TiO_2_, and Zn@TiO_2_), bare cotton fabric, GPTS coated fabric, and functionalized cotton fabrics (i.e., Zn@TiO_2_ coated fabrics) were analyzed by FTIR spectroscopy, and results are presented in Figure 4. Figure 4a presents the FTIR spectra of pristine NPs. The broad band in the range of 3200 cm^−1^ to 3500 cm^−1^ is due to the stretching vibration of the hydroxyl (O-H) group in TiO_2_ and Zn@TiO_2_ NPs. The band at around 1575 cm^−1^ to 1635 cm^−1^ is ascribed to the bending vibrations of coordinated water molecules, in addition to the OH groups associated with Ti–OH. The sharp peak at 775 cm^−1^ in Zn@TiO_2_ NPs and 785 in TiO_2_ NPs is attributed to the M-O vibration. The FTIR spectra of functionalized and uncoated cotton fabric (Figure 4b) exhibited a characteristic broad peak at 3270 cm^−1^, corresponding to OH-stretching vibration due to hydrogen bonding in cellulose [29]. Peaks at 2893 cm^−1^ and 1308 cm^−1^ belong to –CH symmetric stretching and bending vibration bands, respectively [30]. In addition, the absorption band at 1157 cm^−1^ corresponds to (1–4) glycosidic linkages of C–O–C stretching in the cellulose polymer of cotton fiber [31].

Cotton fabric coated only with GPTS show different peaks, representative of characteristic functionalities in cotton. Interestingly, the peak of the epoxy group (in GPTS) at 910 cm^−1^ showed a considerable reduction in intensity (bottom left inset, Figure 4b) after coating on fabric, which is the indication of the additional reaction of epoxy with the hydroxyl group of cellulose. GPTS is used as a binder for the immobilization of Zn@TiO_2_ nanoparticles on cotton fabric, for better durability and stability of functionalized fabrics. The FTIR spectra of functionalized fabrics (bottom right inset, Figure 4b) exhibit characteristics band in the range of 600 cm^−1^–630 cm^−1^, due to the M-O stretching vibration in Zn@TiO_2_ NPs, which is absent in uncoated and GPTS coated fabric, confirming the successful loading of NPs on cotton fabrics.

### 3.2. Photocatalytic Degradation Experimentation

The photocatalytic degradation process of organic pollutants using doped NPs depends on multiple factors, such as solution pH, reaction time, amount of photocatalyst, and amount of doping level. Therefore, the multifactor-dependent process should be optimized in all respects, to achieve the best performance of doped NPs. Analyzing the optimum concentration of dopant in a photocatalyst is of considerable importance. Doping generates a sub-level (heterostructure) in a pristine photocatalyst (i.e., TiO_2_ in this study). The heterostructure provides a trapping site for key radicals, and prevents the recombination of electron-hole pairs to boost photocatalytic activity. Besides prevention in recombination, the addition of a dopant also reduces the energy band gap for the photoexcitation of electrons. However, when the concentration of the dopant exceeds a certain limit, the Zn acts as a recombination center for charge carriers, and lowers the efficiency of the degradation process [32]. Therefore, the reduction in degradation efficiency of photocatalysts at higher Zn doping was observed (Figure 5). Figure 5 showed the UV-visible spectroscopic analysis of MB and MO dyes at various time intervals. The Zn@TiO_2_ with the Zn doping of 0.5 mol percent was the most efficient photocatalyst towards effective dye degradation (for both cationic and anionic dyes). Therefore, Zn@TiO_2_-0.5 was chosen for further study.

All the dyes, namely MB, MO, Y, R, and NB, were subject analysis under different conditions of solution pH (i.e., acidic, basic, neutral). A range of pH was selected (3–11) for MB and MO dye degradation, while Y, R, and NB dye degradation was recorded in acidic (pH = 4) and basic (pH = 9) pH values. All the dye samples (10 ppm) were treated under solar irradiation using Zn@TiO_2_-0.5 (50 mg/100 mL of dye solution) photocatalyst. The light intensity recorded at the time of experimentation in April 2022 was 104,000 ± 2000 lx.

The results are presented in Figure 6a,b. The results showed that, under sunlight, the % degradation of MB dye increases from 82% to 99% when the pH of the solution was changed from 3 to 8. The relationship of change in degradation efficiency of photocatalyst to changes in the solution pH can be elaborated on by considering the TiO_2_ behavior in a variable solution pH. In aqueous media, the TiO_2_ is hydrated to give Ti–OH groups on the surface. In acidic pH, TiO_2_ takes up H^+^ ions and becomes positively charged (Equation (5)). The positively charged TiO_2_ NPs will be an attractive adsorption site for negatively charged species (anionic dye). In a basic environment, TiO_2_ donates protons to HO^-^ ions, and has a negative charge (Equation (6)). MB is a cationic dye, and it will preferentially adsorb on a negatively charged surface. Effective adsorption of dye molecules is mandatory for photocatalysis. Additionally, under acidic conditions, the adsorption of MB on TiO_2_ surfaces will be reduced due to electrostatic repulsion between positively charged photocatalysts and positively charged MB dye molecules resulting in lower degradation efficiency in acidic media [33].
(5)$$\mathrm{In}\;\mathrm{acidic}\;\mathrm{media}\;\;\;\;\;\mathrm{Ti}-\mathrm{OH}+H^{+}⇌{\mathrm{TiOH}}_{2}^{+}$$
(6)$$\mathrm{In}\;\mathrm{basic}\;\mathrm{media}\;\;\;\;\;\mathrm{Ti}-\mathrm{OH}+{\mathrm{HO}}^{-}⇌{\mathrm{TiO}}^{-}+H_{2}O$$

The above-discussed behavior of NPs in acidic and basic conditions of solution pH can be effectively employed to interpret the degradation response of anionic dye in different pH. All the anionic dyes, i.e., MO, Y, R, and NB, respond similarly in the extreme conditions of solution pH. Detailed analyses of MO dye degradation under a range of solution pH (i.e., 2–11) were studied and reported in Figure 5c. Considerable reduction in degradation of MO dye, i.e., 95 to 40%, was observed when the pH of the solution changed from 3 to 9, respectively. The surface chemistry of TiO_2_ NPs in basic media is not supportive of the adsorption of anionic dyes. Therefore, MO dye degradation decreases considerably in basic media. Based on the results of MO dye, the three other dyes were studied under two pH values, i.e., at 3 and 9 (Figure 5d). All the dyes respond similarly to MO; the same justification of such responses can be employed on these dyes [34].

Figure 5e represents the photocatalytic degradation of MB and MO dye using GPTS functionalized TiO_2_ and Zn@TiO_2_(0.5) NPs. In this experiment, the response of GPTS was analyzed in terms of the reduction or increment in the photocatalytic activity of NPs after functionalization with GPTS. The results showed that the functionalization of NPs with GPTS does not reduce the photocatalytic performance of NPs, though a small increment in the response variable was observed in the case of TiO_2_ NPs after functionalization with GPTS. The doped NPs exhibit the almost same performance either with or without the functionalization with GPTS.

### 3.3. K/S (Color Strength)

The results of the K/S analyses are presented in Figure 7. The results showed that the 4 wt% loaded NPs (o.w.f.) exhibited a far better reduction in K/S values as compared to other fabrics. The inset of every bar graph is the real image of fabric (C-Zn@TiO_2_(4) placed in the dark (D) and exposed to sunlight (S) placed for every dye. The results present the far superior photo-catalytically assisted self-cleaning activity of dyed fabrics under sunlight. Interestingly, the extensively used and resistive toward effective degradation textile dyes, i.e., drimaren blue, yellow, and red, also showed excellent degradation under sunlight, ensuring that the functionalized fabrics with Zn@TiO_2_ NPs are excellent candidates for self-cleaning activity under sunlight.

The % reduction in K/S values was calculated using the formula given below:(7)$$\%\;\mathrm{decrease}\;\mathrm{in}\frac{K}{S}=\left(\frac{\left(\frac{K}{S}\right)\mathrm{unexposed}-\left(\frac{K}{S}\right)\mathrm{exposed}}{\left(\frac{K}{S}\right)\mathrm{unexposed}-\left(\frac{K}{S}\right)\mathrm{unstained}}\right)\times 100$$

The values of the percentage reduction in dyes on photo catalytically treated fabrics under sunlight using K/S analysis are presented in Table 1.

### 3.4. Radical Trapping Experiment and Proposed Dye Degradation Mechanism

Photocatalysis of organic pollutants is accompanied by various reactive species, including hydroxyl radicals (${\mathrm{HO}}^{˙}$), electrons ($e^{-}$), holes ($h^{+}$), etc. Therefore, the key radical species responsible for photocatalysis in the present research were estimated using radical trapping scavengers. DMSO (dimethyl sulfoxide) for ${\mathrm{HO}}^{˙}$, EDTA (ethylene-diamine-tetra-acetate) for $h^{+}$, and K_2_Cr_2_O_7_ (potassium dichromate) for $e^{-}$ trapping, were used. A total of 10 mM concentration of each scavenger was used in 100 mL of dye solution [35,36,37]. All the experiments were conducted in ambient sunlight under optimized conditions of solution pH, reaction time, etc.

Figure 8a represents the results of radical trapping scavengers using Zn@TiO_2_-0.5 against MB dyes. Results showed that when no scavenger was added, MB dye degradation was 93%. However, when EDTA was added, dye degradation efficiency decreased from 93% to 67%. EDTA is a $h^{+}$ ions scavenger. 

The possible photocatalytic degradation mechanism of Zn@TiO_2_-0.5 against dyes is discussed below; under sunlight, the electrons become excited from the valence band (VB) to the conduction band (CB) in Zn@TiO_2_ nanoparticles, leaving holes in VB. The holes produced in this way contribute to the hydrolysis of water molecules, and the generation of reactive hydroxyl radicals and $H^{+}$ ions (Figure 8b). The small contribution of the electrons in the dye degradation process may be due to their reaction with the oxygen molecules to form superoxide radicals (Equation (12)). The electron also contributes to the production of hydroxy radicals (Equations (13) and (14)) [38]. The highly reactive hydroxyl radicals effectively degraded the dye molecules into lower molecular weight products (degraded products).
(8)$$\mathrm{Zn}@{\mathrm{TiO}}_{2}-0.5+\mathrm{hv}\to h^{+}+e^{-}$$
(9)$$h^{+}+H_{2}O\to {\mathrm{HO}}^{˙}+H^{+}$$
(10)$$h^{+}+{\mathrm{HO}}^{-}\to {\mathrm{HO}}^{˙}$$
(11)$$h^{+}+\mathrm{dye}\to \mathrm{oxidation}\;\mathrm{products}\;$$
(12)$$e^{-}+O_{2}\to O_{2}^{˙-}$$
(13)$$2O_{2}^{˙-}+2H^{+}\to H_{2}O_{2}+O_{2}$$
(14)$$H_{2}O_{2}+e^{-}\to {\mathrm{HO}}^{˙}+{\mathrm{HO}}^{-}$$
(15)$$e^{-}+\mathrm{dye}\to \mathrm{reduction}\;\mathrm{products}\;$$
(16)$${\mathrm{HO}}^{˙}+\mathrm{dye}+O_{2}\to {\mathrm{CO}}_{2}+H_{2}O+\mathrm{other}\;\mathrm{degradation}\;\mathrm{products}\;$$

### 3.5. Sunlight-Assisted Degradation Potential of Functionalized Zn@TiO_2_ NPs Immobilized on Cotton Substrate against Dispersed Coraline Yellow, Orange, and Red Dyes: The Sublimizable Pollutants

To elucidate the degradation potential of functionalized Zn@TiO_2_ NPs immobilized on cotton fabric against selected disperse dyes, the experiment was run as follows.

The particular amount of three disperse dyes, i.e., coralline yellow, orange, and red, were taken separately and spread on stainless steel substrates. The substrates were heated using a hot plate, and mounted with functionalized fabrics. A similar experiment was performed by taking un-functionalized pristine cotton fabric. All the fabrics were cut into two portions: one batch of each portion was kept in the dark, and the other under sunlight. The sunlight-exposed fabrics were kept under moist conditions to facilitate the degradation process. After 2 h of reaction time, all fabrics were examined for their degradation potential against adsorbed disperse dyes using K/S analysis. The results of K/S analysis were used to calculate the % reduction of dispersed dyes (Table 2.). The results of K/S analysis are presented in Figure 9b.

The results depict considerable adsorption of disperse dyes on both coated and uncoated fabrics (evident by K/S values). Figure 9a represents the almost complete degradation of dyes by functionalized fabrics, as compared to pristine cotton fabrics. The surface chemistry of doped TiO_2_ functionalized NPs facilitates the production of hydroxyl radicals. This promising feature may result in inhibiting the deactivation of photocatalysts caused by the accumulation of deposited carbon (generated during disperse dye degradation) [39].

### 3.6. A Kinetic Study Using Zn@TiO_2_-0.5 against MB and MO Dyes

Figure 10a,c present the linear relations of ln C_o_/C_t_ versus time, and Figure 10b,d presents the linear relationship of (1/C_t_ − 1/C_o_) versus time as first and second order kinetic models, using Zn@TiO_2_-0.5 NPs against MB and MO dyes. The values of K_1_ and K_2_ are calculated from the slope of first and second order kinetics as 0.0087 min^−1^ and 0.0006 L µmol^−1^ min^−1^, respectively. The R^2^ value and rate constant, K, for dye degradation are shown in Table 3. The R^2^ value for first order kinetics is best-fitted relative to the second kinetics model, suggesting the proposed reaction kinetic follows first order model [40].

### 3.7. Antibacterial Activity Analysis and Washing Durability Test

Biological self-cleaning properties of functionalized and pristine cotton fabrics were tested using Staphylococcus (Gram-positive) bacteria, owing to their potential role as an invading bacterium affecting human health. Pristine cotton fabric and C-Zn@TiO_2_-4 before washing and after 10 washes were chosen for this purpose. The selection of functionalized fabric was based on their highest reduction in K/S values. A qualitative antibacterial analysis was performed, and the results are presented in Figure 11. The bacterial counts were made just after contaminating the respective fabric with bacteria (t = 0 h) and 24 h after the incubation period (t = 24 h). The results showed that the pristine cotton fabric was not resistant to bacterial action, as no reduction in cell counts was observed. As a result, the fabric can deteriorate the bacterial action. The functionalized cotton fabrics (C-Zn@TiO_2_(4), without washing) showed excellent antibacterial activity, with a 98% reduction in bacterial counts. Similarly, the fabric after 10 washes (10 W) exhibited a 96% reduction in bacterial growth. Such a small variation in results of bacterial cell reduction by unwashed and washed fabrics is attributed to the effective immobilization of dope NPs using GPTS as a coupling agent. The efficient antibacterial action of functionalized cotton fabric could be attributed to the generation of reactive oxygen species from the catalyst surface that play a key role in bacterial cell wall destruction. The observed minute reduction in the antibacterial activity of 10 washed samples could be due to the reduction in nanoparticle adhesion after multiple cycles of laundry washing. However, the proposed results suggest that the functionalized cotton fabrics exhibited improved antibacterial activity.

The 10 W functionalized fabric was subjected to photocatalytic activity against MB dye, under the reaction conditions described previously (Section 2.4) to check the washing stability of the functionalized fabric. Afterward, the K/S values were obtained, and no considerable increment in color strength was observed by the 10 W fabric (Figure 11c). This suggests the fabric exhibit effective self-cleaning photocatalytic activity after several rounds of washing.

### 3.8. Comfort Properties

#### Air Permeability Water Permeability

The air permeability of fabric refers to the rate of air flowing vertically through a given area of the test specimen, under a specified pressure difference, over a given time [41]. In clothing, air permeability is usually used to determine the ‘breathability’ of various textile fabrics [42]. Therefore, the air and water permeability of pristine and functionalized cotton fabric was checked, and the results are presented in Figure 12a. The pristine fabric exhibited high air permeability. The air permeability of textiles is mainly influenced by the cloth cover factor (i.e., cloth openness), as well as the construction of fabric [43,44]. The high air permeability of uncoated fabric indicated the presence of numerous open holes, which assist the permeation of air through the clothing. Conversely, a very minute decline in air permeability was observed for the functionalized fabric (C-Zn@TiO_2_-4). The decrease in air permeability suggested the effective loading of NPs [45], therefore contributing to the small reduction in air permeability values, i.e., from 42 mm/s to 33 mm/s, which is not a considerable reduction in terms of comfortless of fabric.

Moisture management of textiles is important for the thermal balance of the body [46,47]. Cotton has excellent wettability properties [43,48]. The water permeability results showed no considerable reduction in values, suggesting their comfortability.

Figure 12b presents the tensile strength of uncoated and coated cotton fabric, both without sunlight (WS) and under sunlight (S) exposure for 72 h. The functionalized fabric exhibits clear enhancement in tensile strength, which could be attributed to the chemical and mechanical adhesion of doped nanoparticles to the fibrils, making them capable of bearing extra stress [49]. The final tensile strength of the composite fabric is a function of basic fabric strength and doped nanoparticles. Similarly, the impact of sunlight exposure (for 72 h) on tensile strength is also interesting. No considerable reduction in tensile strength was observed after sunlight exposure. This suggests that the fabric is durable, even after continuous exposure to sunlight. The tensile strength of coated fabric after 32 weeks was also checked. The fabric exhibited a tensile strength value greater than that of pristine cotton fabric. Therefore, this suggests no deterioration of fabric due to coatings, when remained unexposed to sunlight and dry. It is evident from previous studies that immobilization of NPs on cotton fabric results in improved mechanical strength of fabric, as compared to pristine cotton fabrics [50].

## 4. Conclusions

Zn@TiO_2_ nanoparticles immobilized on the cotton fabric were used as an efficient substrate for the photocatalytic degradation of adsorbed pollutants. The NPs were synthesized using the sol–gel method, and refluxed with GPTS as a coupling agent. The functionalized NPs were immobilized on cotton fabric using the dip–dry–cure method. The Zn@TiO_2_ NPs coated fabrics were well characterized using different analytical techniques, and the self-cleaning performance was tested under sunlight. Five commercial dyes as model pollutants, and three disperse reactive dyes as model sublimizable pollutants, were taken to study the sunlight-assisted degradation potential of functionalized fabrics. The efficient photocatalytic activity of Zn-doped functionalized fabric is attributed to the improved charge transfer mechanism of photo-generated charge carriers due to Zn heterojunction. The small contribution of Zn doping (i.e., 0.5 mol%) was found efficient enough for achieving a sunlight-active photocatalytic response of doped NPs. The optimized fabric showed a 90–98% (for five commercial dyes) reduction in dye concentration within 3 h of reaction time under ambient sunlight. The fabrics exhibited excellent stability, as no considerable reduction in photocatalytic activity was observed after 10 washes. This is due to the effective immobilization of doped NPs on fabric, as supported by SEM analysis of 10 washed sample fabrics. Additionally, efficient antibacterial activity was observed by the C-Zn@TiO_2_(4) fabric as a 98% reduction in bacterial cell counts after 24 h of incubation with unwashed, and a 96% reduction with C-Zn@TiO_2_(4) after 10 washes. Therefore, the proposed photocatalytically active self-cleaning fabrics can be used as an efficient candidate for environmental remediation.

## Figures and Tables

**Figure 1 polymers-15-02775-f001:**
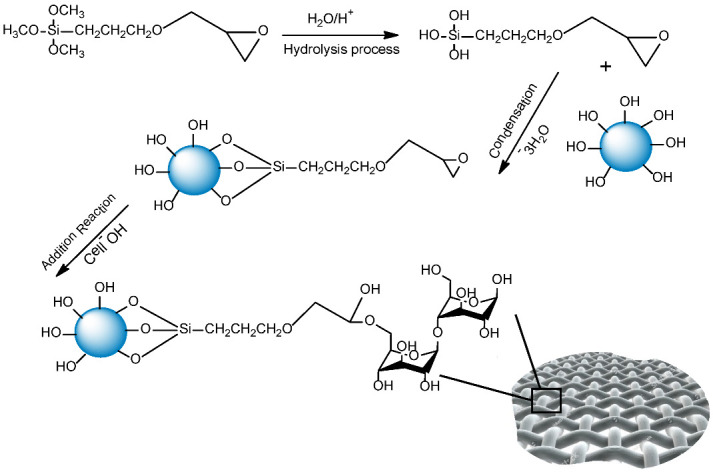
Schematic reaction mechanism for the synthesis and immobilization of TiO_2_ and Functionalized TiO_2_ NPs on cellulosic substrate.

**Figure 2 polymers-15-02775-f002:**
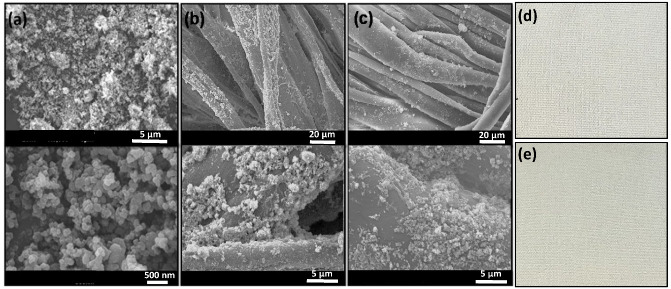
SEM images of (**a**) Zn@TiO_2_-0.5 NPs, (**b**) C-Zn@TiO_2_(4), and (**c**) C-Zn@TiO_2_(4) after 10 washes, (**d**) pristine cotton fabric, and (**e**) functionalized Zn@TiO_2_ coated cotton fabric.

**Figure 3 polymers-15-02775-f003:**
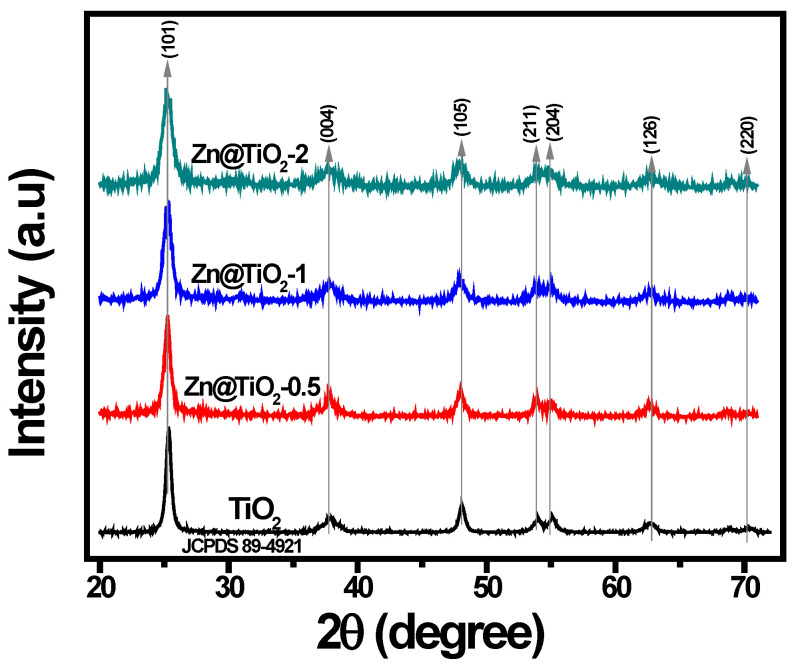
XRD patterns of pristine TiO_2_ and Zn@TiO_2_ NPs.

**Figure 4 polymers-15-02775-f004:**
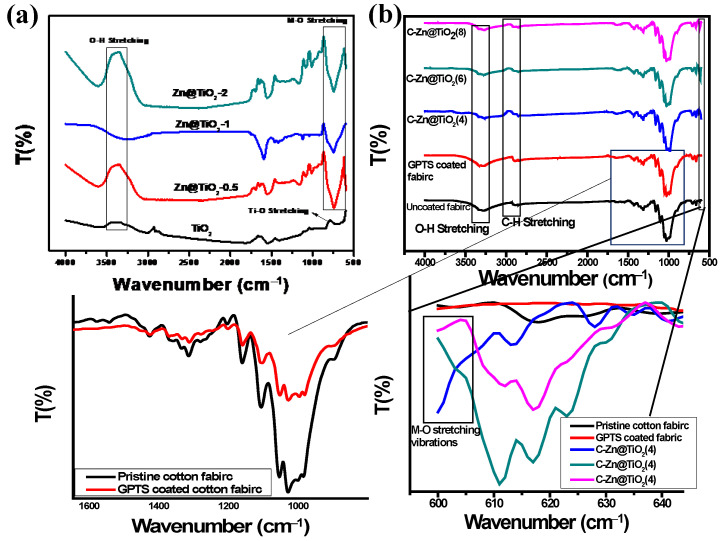
FTIR analysis of (**a**) pristine TiO_2_ and Zn@TiO_2_ NPs, and (**b**) uncoated, GPTS coated, and Zn@TiO_2_ NPs coated fabrics.

**Figure 5 polymers-15-02775-f005:**
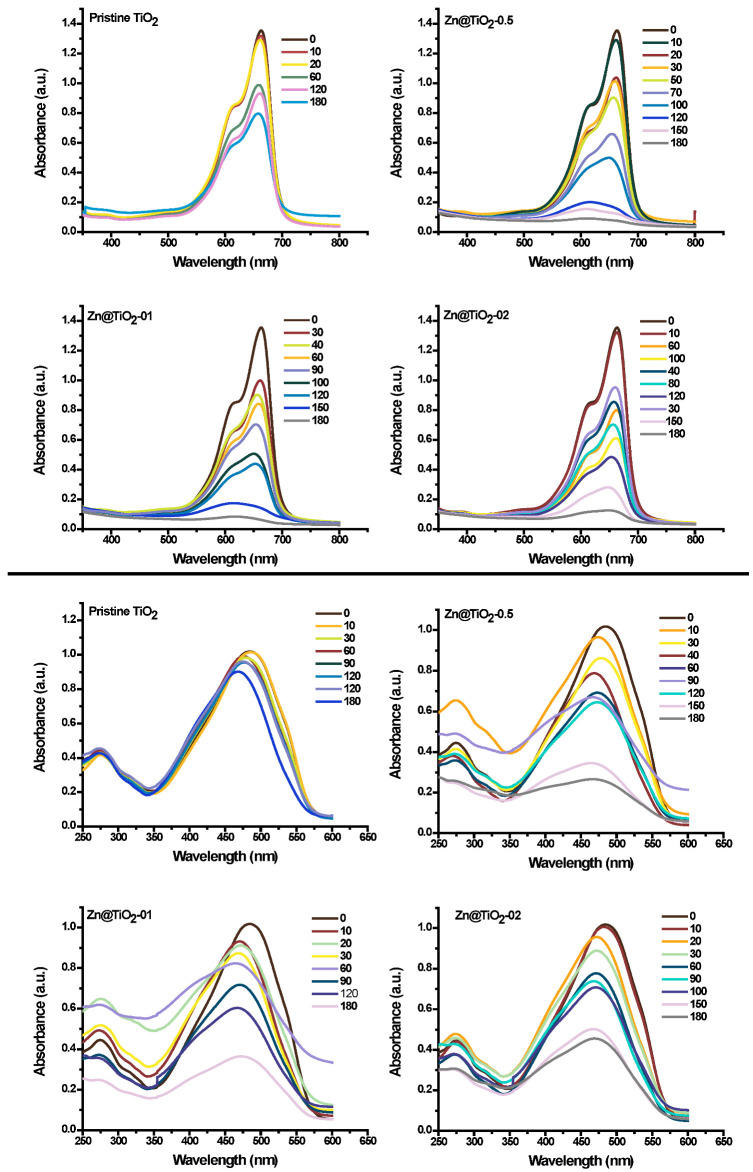
Treatment scans of MB (upper batch) and MO (lower batch), using pristine TiO_2_ NPs and Zn@TiO_2_ NPs with different time intervals (i.e., 0 to 180 min).

**Figure 6 polymers-15-02775-f006:**
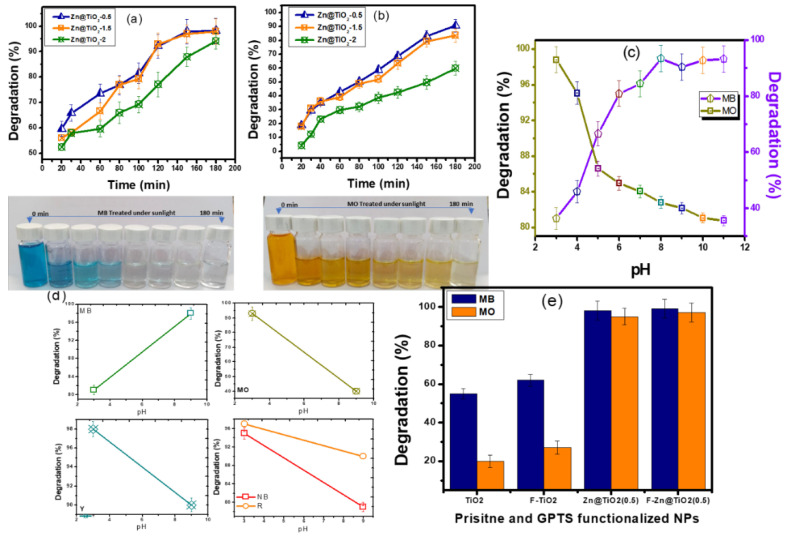
Effect of different doping levels of Zn on % degradation of (**a**) MB and (**b**) MO. Response of Zn@TiO_2_-0.5 (dose = 50 mg/100 mL) (**c**) under different conditions of solution pH using MO and MB, and (**d**) under acidic and basic pH using Y, R, and NB dyes (all dyes concentration = 10 ppm); (**e**) effect of GPTS functionalization on the photocatalytic response of NPs.

**Figure 7 polymers-15-02775-f007:**
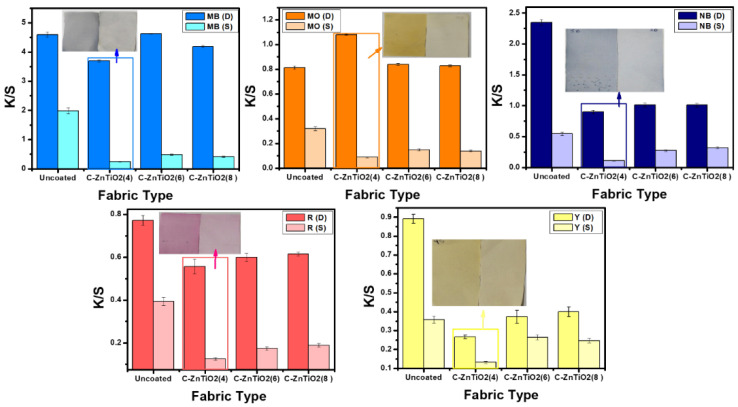
Comparative color strength (K/S) values of dyes loaded uncoated and functionalized fabrics.

**Figure 8 polymers-15-02775-f008:**
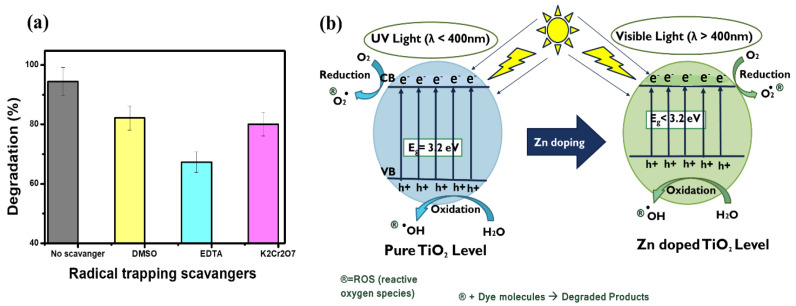
(**a**) Radical scavenging experiment and (**b**) proposed photocatalytic degradation mechanism of pristine and Zn doped TiO_2_ NPs.

**Figure 9 polymers-15-02775-f009:**
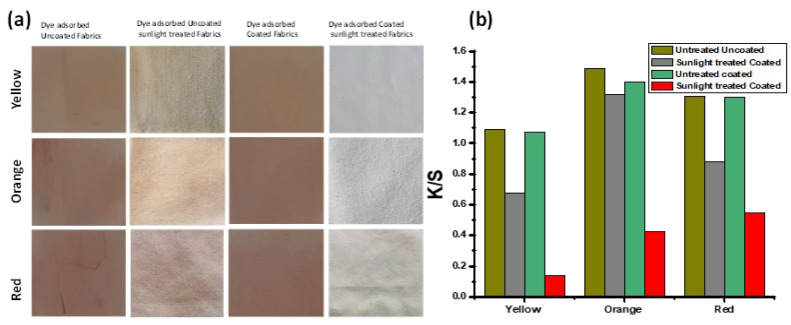
Degradation experiment: (**a**) images of functionalized and pristine fabrics before and after sunlight-assisted photocatalytic degradation of disperse dyes, (**b**) K/S testing of all fabrics.

**Figure 10 polymers-15-02775-f010:**
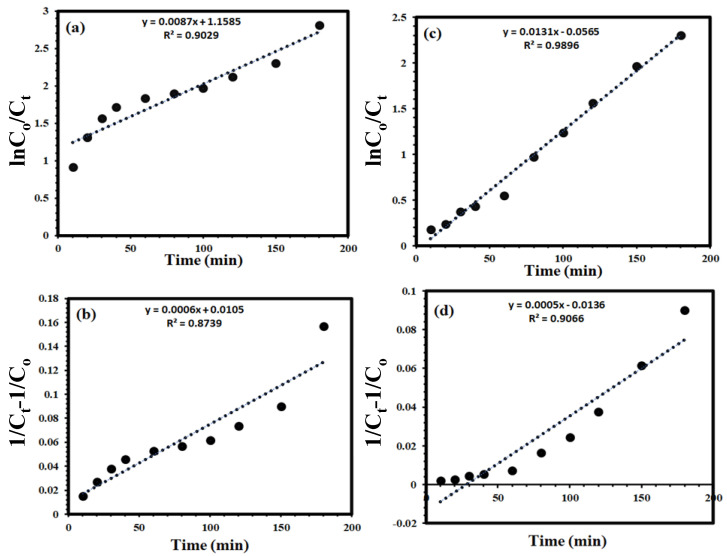
Kinetic study: (**a**,**c**) first order, and (**b**,**d**) second order reaction kinetic models, using MB (**a**,**b**) and MO (**c**,**d**) dyes.

**Figure 11 polymers-15-02775-f011:**
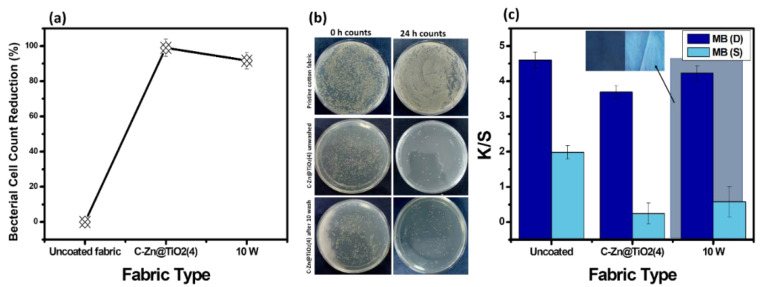
Antibacterial Activity (**a**,**b**) and washing durability test (**c**).

**Figure 12 polymers-15-02775-f012:**
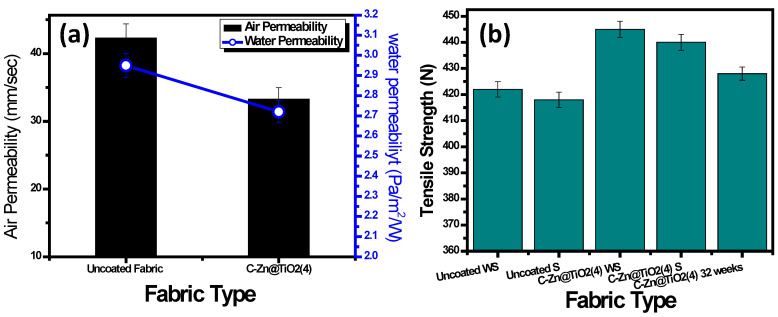
(**a**) Air permeability and water permeability analysis. (**b**) Tensile strength analysis of uncoated and functionalized fabrics.

**Table 1 polymers-15-02775-t001:** % reduction of dyes using K/S values.

Fabric Type	Dye	% Reduction	Fabric Type	Dye	% Reduction
C-Zn@TiO_2_(04)	MB	95	C-Zn@TiO_2_(08)	MB	91
	MO	97		MO	90
	Y	67		Y	44
	R	88		R	77
	NB	93		NB	72
C-Zn@TiO_2_(06)	MB	90	Uncoated	MB	35
	MO	89		MO	39
	Y	35		Y	28
	R	79		R	39
	NB	72		NB	35

**Table 2 polymers-15-02775-t002:** Percentage reduction in disperse dyes obtained by K/S values.

Disperse Dyes	Fabric Type	
	Uncoated	C-Zn@TiO_2_(4)
Yellow	39.25	88.73
Orange	11.18	70.5
Red	33.44	58.5

**Table 3 polymers-15-02775-t003:** The calculated reaction rate constants of Zn@TiO_2_-0.5 towards photodegradation of MB and MO dye.

Dye	1st Order Kinetics		2nd Order Kinetics	
	R^2^	K_1_ (min^−1^)	R^2^	K_2_ (L µmol^−1^min^−1^)
MB	0.90	0.0087	0.87	0.0006
MO	0.99	0.0131	0.91	0.0005

## Data Availability

Not applicable.

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
