# Peer review of "Sunlight-Driven Photocatalytic Active Fabrics through Immobilization of Functionalized Doped Titania Nanoparticles"

_polymers, 2023, doi:10.3390/polym15132775_

Round 1
Reviewer 1 Report
In this manuscript, the authors developed cotton fabrics with properties of wastewater treatment and VOCs degradation by immobilizing functionalized Zn-doped TiO2 nanoparticles. The functional cotton fabrics are endowed with excellent pollutant degradation performance while maintaining good mechanical properties and comfort. Although this work has achieved good results, there are still some problems to be addressed before accepting for publication.
1. According to the SEM images of the fabrics, a layer of Zn-doped TiO2 nanoparticles was attached to the surface of the fabrics after functional modification. Macroscopically, is there any difference between the modified fabrics and the original fabrics? Relevant photos should be provided to show their macroscopic appearance.
2. From the durability test results of the fabrics, it can be seen that the degradation performance of the functional fabrics to pollutants changes after 10 times of washing. Is there any change in the microstructure of the fabric? The authors should provide relevant data.
3. Figure 4 shows the degradation performance of different Zn-doped TiO2 nanoparticles to organic matters at different times. The authors should mark the corresponding time of each absorbance curve clearly.
4. There are a lot of mistakes in this manuscript. For example, the spelling of Celsius on page 4 is wrong; the article contains a large number of "Error! Reference source not found ", etc. The author should read through the whole manuscript carefully and make corrections.
The language needs to be improved appropriately.
Author Response
Comments and Suggestions for Authors
In this manuscript, the authors developed cotton fabrics with properties of wastewater treatment and VOCs degradation by immobilizing functionalized Zn-doped TiO2 nanoparticles. The functional cotton fabrics are endowed with excellent pollutant degradation performance while maintaining good mechanical properties and comfort. Although this work has achieved good results, there are still some problems to be addressed before accepting for publication.
- According to the SEM images of the fabrics, a layer of Zn-doped TiO2 nanoparticles was attached to the surface of the fabrics after functional modification. Macroscopically, is there any difference between the modified fabrics and the original fabrics? Relevant photos should be provided to show their macroscopic appearance.
Response: The macroscopic pictures of pristine and functionalized Zn@TiO2 NPs immobilized cotton fabrics are added into the updated manuscript Figure 3 part d and e. There is no apparent change in the physical properties of functionalized NPs coated fabircs.
- From the durability test results of the fabrics, it can be seen that the degradation performance of the functional fabrics to pollutants changes after 10 times of washing. Is there any change in the microstructure of the fabric? The authors should provide relevant data.
Response: The microscopic image (SEM results in Figure 1c) represents the fabric structure after 10 washes. The high-resolution image demonstrates almost the same distribution of NPs on the washed sample as on without washed sample. This represents the strong immobilization of GPTS functionalized NPs on textile substrates. The little reduction in the degradation performance of fabric after 10 washes may be attributed to the little washing of adsorbed NPs. However, the reduction in efficiency is still encouraging.
- Figure 4 shows the degradation performance of different Zn-doped TiO2 nanoparticles to organic matters at different times. The authors should mark the corresponding time of each absorbance curve clearly.
Response: Thank you for your notice, Figure 4 has been updated accordingly.
- There are a lot of mistakes in this manuscript. For example, the spelling of Celsius on page 4 is wrong; the article contains a large number of "Error! Reference source not found ", etc. The author should read through the whole manuscript carefully and make corrections.
Response: The manuscript has been updated thoroughly to eradicate any typos or grammatical errors. Some of the corrections are highlighted in the updated version.

Reviewer 2 Report
This paper reports the modification of textile substrates with Zn-doped TiO2 nanoparticles to impart self-cleaning property. The photocatalytic activity of the resulting products towards dyes and VOC has been demonstrated, and the coated textiles exhibit bactericidal property. The study is interesting; however, there is a severe problem with this approach. The textiles should be exposed to sunlight, and polymer fibers can also be degraded if the photo-catalytically active species are present on their surface. The authors should quantify the light stability of the modified textiles. This is especially relevant for dyed textiles, while fading during light exposure can be a big problem.
In many places it is shown "Error! Reference source not found". This should certainly be corrected.
Author Response
This paper reports the modification of textile substrates with Zn-doped TiO2 nanoparticles to impart self-cleaning property. The photocatalytic activity of the resulting products towards dyes and VOC has been demonstrated, and the coated textiles exhibit bactericidal property. The study is interesting; however, there is a severe problem with this approach. The textiles should be exposed to sunlight, and polymer fibers can also be degraded if the photo-catalytically active species are present on their surface. The authors should quantify the light stability of the modified textiles. This is especially relevant for dyed textiles, while fading during light exposure can be a big problem.
Response: Thank you for appreciating the work. As far as the degradation of fibers is concerned by the generation of hydroxyl radicles, we agree up to some extent. The only possibility for the generation of reactive radicles is during sunlight exposure with a considerable humidity factor. In usual wearing conditions, it is a rare scenario. Right after washing the clothes, the situation may exist which supports the generation of reactive radicles but only during sunlight exposure.
We have already presented the fabric tensile strength results after exposure to sunlight (continuous) for 72 hr while wetting the fabric continuously (Figure 11b). In routine washing, the time required for sunlight exposure is usually 1 to 2 hr. The results showed no considerable reduction in T.S measurements after 72 hr of direct and continuous sunlight exposure. Which represents the promising T.S of fabric. In the revised manuscript the tensile strength of coated fabric after almost 32 weeks is presented. The fabric still exhibits a tensile strength value greater than that of uncoated cotton fabric. Please see the updated Figure 11 b.
The proposed work was studied on undyed cotton fabric. Therefore, no measurements or observations were made for dyed fabrics. It could be considered in our future work.
Comments on the Quality of English Language
In many places it is shown "Error! Reference source not found". This should certainly be corrected.
Response: No error has been found

Reviewer 3 Report
In this report, Arfa et al. describe the covalent attachment of Zn-doped titanium dioxide particles to cotton fabric to impart photocatalytic self-cleaning property. The level of doping is varied as is the loading on the fabric. These materials are characterized by SEM, FTIR, XRD, color strength, bactericidal properties and degradation of selected dyes and VOCs.
The findings in this paper present an interesting application of the self-cleaning capabilities with incorporated TiO2, which is perhaps best known for use in paints. At suitable values of pH, high degradation rates have been found for the model dye compounds using the doped TiO2. The effect of pH on extent of degradation is attributed to change in the surface charge of the particles; a reference value for the point of zero charge might be added. Data also supports the degradation of the dyes on the modified fabric.
The research in this submission is significant and merits publication. On the other hand, there are major issues with the writing of the manuscript and how the authors present the application. The authors have not proven in fact that the fabric has been cleaned. It is more likely that the “soiled” cotton has simply been bleached with degradation products remaining. This is an important question to be addressed by the authors since washing may still result in a contaminated stream with harmful effluents.
Work is needed on a coherent description of the purpose of the research in the title, abstract and introduction. The words in the title “cleaning of wastewater” suggests the paper will be about a process for active removal by treatment of wastewater. However, the testing focused on air-dried fabric samples with “cleaning” taking place on the textile. The implication is that cotton will be cleaned as worn out-of-doors.
The study should be discussed in the context of prior related work.
Nanoparticles are generally considered to be smaller than 100 nm. It is not clear from the SEM that the titanium dioxide particles fall in this size range. Provide a size distribution of the particles.
Add some basic description of the cotton fabric used.
Experimental techniques should be placed in the Materials and Methods section, not the Results and Discussion section.
Disperse Yellow, Disperse Orange and Disperse Red have high boiling. They are not volatile compounds and unsuited to be used as VOCs in experiments.
Sections of the text cannot be understood with “Error! Reference source not found.”
Add a key to Figures 4. First describe the difference in degradation rate with the level of doping and then explain. Move lines 297 to 300. Figure 5a is not referenced in the text.
Chemical formulas for titanium dioxide and zinc nitrate hexahydrate should use proper notation.
Errors in superscript for degrees Celsius.
Many capitalization and subscript errors for rate constants and K/S
Capitalization and omission errors for GPTS
Units for light intensity are “lx”, not LUX
Use consistent capitalization for o.w.f.
Capitalization errors in SIGMA ALDRICH, DAEJUNG, CECIL Instrument, LUX. Zinc, Uncoated
some of the reaction in section 3.7 need to be balanced
Subscripts for Co and Ct.
Line 28: The abstract indicates elemental compositions were obtained. Results are missing?
Line 156: (4, 6, and 8%)
Line 188: It has been suggested that the so-called Debye-Scherrer equation should simply be called the Scherrer equation(“The Scherrer-Equation versus the Debye-Scherrer Equation”, Nature Nanotechnol., 2011)
Line 193: GretagMacbeth
Line 231: nondimensional? The meaning is not clear. “….evenly shaped”?
Line 238: Would “apparent” be better than “potential”?
Line 247: Superscript ion charge
Line 267: spectra
Line 286: fabrics
Line 315: Figures 5a-c?
Line 352: sentence fragment
Lines 412-13: Adsorption is generally an equilibrium process. Explain what is meant by VOCs being re-emitted when the surface is saturated.
Line 418: verb-subject
Line 434: radicals; Sentence is unclear.
Lines 539-545: The first author appears to have made no contributions to this paper!
The errors in language dealt mostly with capitalization, subscripts and punctuation with less frequent issues in sentences fragments and verb-subject agreement. The number of errors overall suggest a cavalier attitude of the 7 authors in proofreading their manuscript.
Author Response
Comments and Suggestions for Authors
In this manuscript, the authors developed cotton fabrics with properties of wastewater treatment and VOCs degradation by immobilizing functionalized Zn-doped TiO2 nanoparticles. The functional cotton fabrics are endowed with excellent pollutant degradation performance while maintaining good mechanical properties and comfort. Although this work has achieved good results, there are still some problems to be addressed before accepting for publication.
- According to the SEM images of the fabrics, a layer of Zn-doped TiO2 nanoparticles was attached to the surface of the fabrics after functional modification. Macroscopically, is there any difference between the modified fabrics and the original fabrics? Relevant photos should be provided to show their macroscopic appearance.
Response: The macroscopic pictures of pristine and functionalized Zn@TiO2 NPs immobilized cotton fabrics are added into the updated manuscript Figure 3 part d and e. There is no apparent change in the physical properties of functionalized NPs coated fabircs.
- From the durability test results of the fabrics, it can be seen that the degradation performance of the functional fabrics to pollutants changes after 10 times of washing. Is there any change in the microstructure of the fabric? The authors should provide relevant data.
Response: The microscopic image (SEM results in Figure 1c) represents the fabric structure after 10 washes. The high-resolution image demonstrates almost the same distribution of NPs on the washed sample as on without washed sample. This represents the strong immobilization of GPTS functionalized NPs on textile substrates. The little reduction in the degradation performance of fabric after 10 washes may be attributed to the little washing of adsorbed NPs. However, the reduction in efficiency is still encouraging.
- Figure 4 shows the degradation performance of different Zn-doped TiO2 nanoparticles to organic matters at different times. The authors should mark the corresponding time of each absorbance curve clearly.
Response: Thank you for your notice, Figure 4 has been updated accordingly.
- There are a lot of mistakes in this manuscript. For example, the spelling of Celsius on page 4 is wrong; the article contains a large number of "Error! Reference source not found ", etc. The author should read through the whole manuscript carefully and make corrections.
Response: The manuscript has been updated thoroughly to eradicate any typos or grammatical errors. Some of the corrections are highlighted in the updated version.
Reviewer # 2
Comments and Suggestions for Authors
In this report, Arfa et al. describe the covalent attachment of Zn-doped titanium dioxide particles to cotton fabric to impart photocatalytic self-cleaning property. The level of doping is varied as is the loading on the fabric. These materials are characterized by SEM, FTIR, XRD, color strength, bactericidal properties and degradation of selected dyes and VOCs.
The findings in this paper present an interesting application of the self-cleaning capabilities with incorporated TiO2, which is perhaps best known for use in paints. At suitable values of pH, high degradation rates have been found for the model dye compounds using the doped TiO2. The effect of pH on extent of degradation is attributed to change in the surface charge of the particles; a reference value for the point of zero charge might be added. Data also supports the degradation of the dyes on the modified fabric.
The research in this submission is significant and merits publication. On the other hand, there are major issues with the writing of the manuscript and how the authors present the application. The authors have not proven in fact that the fabric has been cleaned. It is more likely that the “soiled” cotton has simply been bleached with degradation products remaining. This is an important question to be addressed by the authors since washing may still result in a contaminated stream with harmful effluents.
Response: Thank you for your comment. Authors regret the missing data on cotton fabric and its prior treatment before coating and application. The manuscript is updated as per reviewers comment. The Mercerized cotton fabric was used for the present study and pretreatment was given to facilitate the adsorption of functionalized NPs on cotton fabric. The detail of the fabric and its pretreatment is mentioned in sections 2.1 and 2.2.4. however washing
Work is needed on a coherent description of the purpose of the research in the title, abstract, and introduction. The words in the title “cleaning of wastewater” suggests the paper will be about a process for active removal by treatment of wastewater. However, the testing focused on air-dried fabric samples with “cleaning” taking place on the textile. The implication is that cotton will be cleaned as worn out-of-doors.
Response: Thank you. The need of research and manuscript theme is mentioned in abstract (Highlighted) and in the last paragraph of introduction. Further, the title has been revised as per suggestion. The testing of fabric was performed while the fabric was kept under moist conditions. The fabric performs photocatalytic under moist conditions.
The study should be discussed in the context of prior related work.
Response: The new citation referring to the context of present research has been added. Please see section 3.8.
Nanoparticles are generally considered to be smaller than 100 nm. It is not clear from the SEM that the titanium dioxide particles fall in this size range. Provide a size distribution of the particles.
Response: The nanoparticle size has been calculated using image j software and the resultant size distribution is labeled for your reference below. Majority of the nanoparticles are below 100 nm. The agglomeration of nanometer-ranged particles has also been observed.
Add some basic description of the cotton fabric used.
Response: The description of the cotton fabric used has been added to section 2.1.
Experimental techniques should be placed in the Materials and Methods section, not the Results and Discussion section.
Response: Dear reviewer, the experimental techniques used for the present study is already presented in the Material and Methods section under the heading of section 2.3 Testing and characterization
Disperse Yellow, Disperse Orange, and Disperse Red have high boiling. They are not volatile compounds and unsuited to be used as VOCs in experiments.
Response: The dispersed dyes are mostly used for sublimation printing. Considering their sublimation properties, these dyes were selected as volatile organic compounds for their degradation study.
Sections of the text cannot be understood with “Error! Reference source not found.”
Response: Text updated.
Add a key to Figure 4. First describe the difference in degradation rate with the level of doping and then explain. Move lines 297 to 300. Figure 5a is not referenced in the text.
Response: The key to Figure 4 has been added. Figure 5 a referred to on line 317.
Chemical formulas for titanium dioxide and zinc nitrate hexahydrate should use proper notation.
Errors in superscript for degrees Celsius.
Many capitalization and subscript errors for rate constants and K/S
Capitalization and omission errors for GPTS
Units for light intensity are “lx”, not LUX
Use consistent capitalization for o.w.f.
Capitalization errors in SIGMA ALDRICH, DAEJUNG, CECIL Instrument, LUX. Zinc, Uncoated
some of the reactions in section 3.7 need to be balanced
Response: All the pointed errors have been eradicated and highlighted in the updated manuscript. Besides the balancing of equations has been checked.
Subscripts for Co and Ct.
Response: Subscriptions considered.
Line 28: The abstract indicates elemental compositions were obtained. Results are missing?
Response: Thank you for your notice. The sentence in the abstract is updated now.
Line 156: (4, 6, and 8%)
Response: Correction has been made.
Line 188: It has been suggested that the so-called Debye-Scherrer equation should simply be called the Scherrer equation(“The Scherrer-Equation versus the Debye-Scherrer Equation”, Nature Nanotechnol., 2011)
Response: Suggestion has been considered.
Line 193: GretagMacbeth
Response: Correction has been made.
Line 231: nondimensional? The meaning is not clear. “….evenly shaped”?
Response: The sentence has been restructured.
Line 238: Would “apparent” be better than “potential”?
Response: The suggestion has been considered
Line 247: Superscript ion charge
Response: Correction has been made
Line 267: spectra
Response: Correction has been made
Line 286: fabrics
Response: Correction has been made
Line 315: Figures 5a-c?
Response: We have noticed the typo error is describing the Figure 5 a and c in text. The correction has been made in updated manuscript.
Line 352: sentence fragment
Response: Correction has been made.
Lines 412-13: Adsorption is generally an equilibrium process. Explain what is meant by VOCs being re-emitted when the surface is saturated.
Response: During the adsorption process the particles are in contact with the surface layer of the adsorbent. While in multilayer adsorption more than one layer of adsorbates formed. The multilayer adsorption process is physical adsorption which requires less energy to be released again into the environment as compared to the single layer adsorption (chemical adsorption) which detached only on degradation. Therefore, once the surface is saturated there is more chance for the release of VOCs into the environment.
Line 418: verb-subject
Response: Correction has been made and highlighted
Line 434: radicals; Sentence is unclear.
Response: When Zn@TiO2 is triggered by light absorption, the prevention of recombination of charge carriers is promoted (due to heterojunction created by Zn doping). Thereby facilitating the generation of hydroxyl radicles by redox reactions. The as-produced OH radicles decompose/degrade the adsorbed pollutants and the continuity of this process prevents the deactivation of the catalyst.
Lines 539-545: The first author appears to have made no contributions to this paper!
Response: The author's name is Umme Arfa (U.A) in the previously version the name was mistakenly written as Arfa Maqbool (A.M). Thank you for your notice the manuscript has been updated now.
Comments on the Quality of English Language
The errors in language dealt mostly with capitalization, subscripts, and punctuation with less frequent issues in sentences fragments and verb-subject agreement. The number of errors overall suggest a cavalier attitude of the 7 authors in proofreading their manuscript.
Response: The manuscript has been updated thoroughly to eradicate any typos or grammatical errors. Some of the corrections are highlighted in the updated version.

Round 2
Reviewer 3 Report
The manuscript is much improved. It represents a comprehensive study with analysis of experimental findings. There remain some points for the authors to consider.
No response was provided to the following comment: “The authors have not proven in fact that the fabric has been cleaned. It is more likely that the “soiled” cotton has simply been bleached with degradation products remaining. This is an important question to be addressed by the authors since washing may still result in a contaminated stream with harmful effluents.” There is a difference between actual removal of molecules(“cleaning”) and creating the appearance of its removal(“bleaching”). We can only say that the chromophore has been altered. In the discussion, the authors might add a few sentences making it clear to the reader that they recognize the difference. Bleaching generally makes soiling species more soluble, perhaps with less need for surfactants in line with their objectives.
Results in Table 1 and elsewhere demonstrate that the modified fabrics can degrade dyes. The authors have effectively established that dyes can be degraded. It is not clear how the so-called VOC experiments add anything beyond the findings with MO, MB and the other dyes. As noted previously, the dispersed dyes are not VOCs. The authors need to explain the difference to the reader in the manuscript and justify why these particular nonvolatile compounds are acceptable for proving effectiveness in treating VOCs.
Experimental methods remain in the Results and Discussion section. Here is an example from Section 3.3: “The color strength of uncoated and coated fabrics with variable percentages of NPs was tested by dipping the respective cotton fabric into the dye solution (50 ppm for all dyes). After dipping, one portion of the fabric was placed in the dark, and 2nd portion was placed under ambient sunlight to observe the self-cleaning activity of functionalized fabrics. After exposure for 3 h, the fabrics were subjected to K/S analysis.”
Line 134: mercerized
Line 227 and Table 3: k1 and k2
Line 293: experimentation
Line 438: This sentence is still unclear. Look at sentence structure. “radicals”, not radicle which is a technical term from botany related to seed germination
Equation 7: K/S (4 times)
Reactions 13 and 14 remain unbalanced.
Overall, the English of the revision is fine. However, there were far too many errors in the original submission that should not have been left for reviewers and editors to find.
Author Response
Comments and Suggestions for Authors
The manuscript is much improved. It represents a comprehensive study with analysis of experimental findings. There remain some points for the authors to consider.
No response was provided to the following comment: “The authors have not proven in fact that the fabric has been cleaned. It is more likely that the “soiled” cotton has simply been bleached with degradation products remaining. This is an important question to be addressed by the authors since washing may still result in a contaminated stream with harmful effluents.” There is a difference between actual removal of molecules(“cleaning”) and creating the appearance of its removal(“bleaching”). We can only say that the chromophore has been altered. In the discussion, the authors might add a few sentences making it clear to the reader that they recognize the difference. Bleaching generally makes soiling species more soluble, perhaps with less need for surfactants in line with their objectives.
Thank you for raising an important aspect of study. The concept of photocatalytically active fabric is aligned with the degradation of pollutants into mineralized products on the surface of functionalized fabrics under humid conditions. In literature, there are reports about the mineralization of adsorbed pollutants followed by the degradation of large pollutant molecules into smaller ones and ultimately mineralized ones. For example, Li and coworkers (http://dx.doi.org/10.4028/www.scientific.net/MSF.866.171) reported the mineralization of dye molecules using simple TiO2 immobilized textiles by GC-MS analysis. Whereas in the proposed research, the photocatalytic performance of fabrics is studied using Zn-doped TiO2 NPs, which is more effective as compared to the pristine TiO2 coated fabrics. Therefore, it is inferred that the process is not the decolorization of dyes but it is photocatalytic degradation of adsorbed pollutants into degraded and mineralized products under specific reaction conditions. All the reaction conditions have been optimized and presented. However, the authors agree that comprehensive support for this discussion in the form of GC-MS is highly encouraged. This approach could be accessed in the future study.
Results in Table 1 and elsewhere demonstrate that the modified fabrics can degrade dyes. The authors have effectively established that dyes can be degraded. It is not clear how the so-called VOC experiments add anything beyond the findings with MO, MB and the other dyes. As noted previously, the dispersed dyes are not VOCs. The authors need to explain the difference to the reader in the manuscript and justify why these particular nonvolatile compounds are acceptable for proving effectiveness in treating VOCs.
The authors agreed with the reviewer's point that disperse dyes should not be treated as the VOC's representatives. The dispersed dyes sublimate when heated at a temperature between 150 - 230 °C. Considering their sublimation property, the dispersed dyes were selected and said as VOCs representatives. However, the authors will consider the degradation of compounds like benzene, ethylene glycol, formaldehyde, methylene chloride, etc as true representatives of VOCs. That is why in the updated manuscript, the concept of VOCs has been removed. Thank you for your comment.
Experimental methods remain in the Results and Discussion section. Here is an example from Section 3.3: “The color strength of uncoated and coated fabrics with variable percentages of NPs was tested by dipping the respective cotton fabric into the dye solution (50 ppm for all dyes). After dipping, one portion of the fabric was placed in the dark, and 2nd portion was placed under ambient sunlight to observe the self-cleaning activity of functionalized fabrics. After exposure for 3 h, the fabrics were subjected to K/S analysis.”
The data has been moved to the experiment section under the new heading of 2.5.
Line 134: mercerized
The correction has been made
Line 227 and Table 3: k1 and k2
The correction has been made
Line 293: experimentation
Line 438: This sentence is still unclear. Look at sentence structure. “radicals”, not radicle which is a technical term from botany related to seed germination
The correction has been made
Equation 7: K/S (4 times)
The correction has been made
Reactions 13 and 14 remain unbalanced.
Equations are balanced now. Thank you for your valuable comments in improving the quality of the manuscript. .
